# Influence of Roasting Temperature on the Detectability of Potentially Allergenic Lupin by SDS-PAGE, ELISAs, LC-MS/MS, and Real-Time PCR

**DOI:** 10.3390/foods13050673

**Published:** 2024-02-23

**Authors:** Bruno Beyer, Dominik Obrist, Philipp Czarda, Katharina Pühringer, Filip Vymyslicky, Barbara Siegmund, Stefano D’Amico, Margit Cichna-Markl

**Affiliations:** 1Department of Analytical Chemistry, Faculty of Chemistry, University of Vienna, Währinger Straße 38, 1090 Vienna, Austria; a01615852@unet.univie.ac.at (B.B.); a11913380@unet.univie.ac.at (D.O.); a11771735@unet.univie.ac.at (P.C.); katharina.puehringer@univie.ac.at (K.P.); 2Doctoral School in Chemistry, University of Vienna, Währinger Str. 38-40, 1090 Vienna, Austria; 3Department of Analytical Chemistry, Faculty of Science, Charles University, Hlavova 8/2030, 12843 Prague 2, Czech Republic; f.vymyslicky@gmail.com; 4Institute of Analytical Chemistry and Food Chemistry, Graz University of Technology, Stremayrgasse 9/II, 8010 Graz, Austria; barbara.siegmund@tugraz.at; 5AGES—Austrian Agency for Health and Food Safety, Institute for Animal Nutrition and Feed, Spargelfeldstrasse 191, 1220 Vienna, Austria; stefano.d-amico@ages.at

**Keywords:** lupin, *Lupinus angustifolius*, cultivar Boregine, food allergen, roasting, detectability, SDS-PAGE, ELISA, LC-MS, real-time PCR

## Abstract

Seeds of “sweet lupins” have been playing an increasing role in the food industry. Lupin proteins may be used for producing a variety of foods, including pasta, bread, cookies, dairy products, and coffee substitutes. In a small percentage of the population, lupin consumption may elicit allergic reactions, either due to primary sensitization to lupin or due to cross-allergy with other legumes. Thus, lupin has to be declared on commercial food products according to EU food regulations. In this study, we investigated the influence of roasting seeds of the *L. angustifolius* cultivar “Boregine” on the detectability of lupin by sodium dodecyl sulfate-polyacrylamide gel electrophoresis (SDS-PAGE), ELISAs, LC-MS/MS, and real-time PCR. Seeds were roasted by fluidized bed roasting, and samples were drawn at seed surface temperatures ranging from 98 °C to 242 °C. With increasing roasting temperature, the extractability of proteins and DNA decreased. In addition, roasting resulted in lower detectability of lupin proteins by ELISAs and LC-MS/MS and lower detectability of DNA by real-time PCR. Our results suggest reduced allergenicity of roasted lupin seeds used for the production of “lupin coffee”; however, this has to be confirmed in in vivo studies.

## 1. Introduction

The percentage of consumers purchasing plant-based foods has been increasing tremendously in recent years. Consumers are opting for plant-based food products mainly for health, ethical, and/or ecological reasons [1]. Plant-based meat alternatives frequently contain proteins from legumes, including soy, pea, lentil, chickpea, and lupin [2]. Among more than 450 species of the *Lupinus* genus, only seeds of special breeds with reduced quinolizidine alkaloid content, so-called “sweet lupins”, are used for food production [3]. The lupin species *L. albus* (white lupin), *L. angustifolius* (blue or narrow-leaf lupin), *L. luteus* (yellow lupin), and *L. mutabilis* (Andean lupin) are currently playing the most important role in the food industry.

Lupin seeds are rich in proteins, comparable to soybean seeds, and dietary fiber [4]. Due to the rather high content of essential amino acids, fortification of foods with lupin proteins not only increases their protein content but also their nutritional value [5]. Consumption of lupin seeds has been associated with a variety of health benefits, including hypoglycemic, hypocholesterolemic, and anti-inflammatory effects [5,6,7,8]. Since lupin protein isolates and concentrates show outstanding techno-functional properties, they are used to increase, e.g., water holding, emulsifying, and/or gelation capacity of food products [9,10].

Lupin proteins may be found in various food products, including pasta, bread, cookies, and dairy products [9]. Coffee substitutes made from roasted lupin seeds (“lupin coffee”) is a speciality, appreciated by consumers because of its mild, slightly nutty taste. “Lupin coffee” is not only a naturally caffeine-free alternative to coffee prepared from coffee beans, but also a climate-friendly one, because lupin plants can be grown in low fertility soils in various climate zones [11].

Since lupin seeds are inherently gluten-free, lupin-based food products are well tolerated by persons suffering from celiac disease. However, lupin consumption may trigger allergic symptoms in a small percentage of the population. Allergic symptoms are a consequence of either primary sensitization to lupin or cross-allergy with other legumes [12]. Due to sequence homology and the occurrence of common epitopes in legume seed proteins, cross-allergy predominantly occurs with peanuts, but has also been reported for soybeans, lentils, beans, chickpeas, and peas [13,14]. In general, seed storage proteins are classified into four fractions, 11S globulin (conglutin α), 7S globulin (conglutin β, vicilins), 7S basic globulin (conglutin γ), and 2S sulfur-rich albumin (conglutin δ) [15]. In lupin seeds, the conglutin α and conglutin β content surpasses that of conglutin γ and conglutin δ. However, exact concentration and the ratio of conglutins differ between lupin species and cultivars and are influenced by environmental factors as well [16]. Several allergenic proteins have already been identified in lupin seeds [17,18]. To date, the World Health Organization (WHO)/International Union of Immunological Societies (IUIS) allergen nomenclature subcommittee and the ALLERGOME (http://www.allergome.org/, accessed on 15 June 2023) databases assign different families of lupin proteins as allergens, in detail, globulins (α-, β-, and γ-conglutin), 2S albumins (δ conglutin), and a few minor fractions such as nonspecific lipid transfer proteins (nsLTP) (Lup an 3), pathogenesis-related (PR)-10 proteins (Lup a 4 and Lup l 4), and profilins (Lup a 5) [19]. Studies have shown the presence of several important IgE-reactive proteins in lupin, of which β-conglutin was identified as the major allergen in *L. angustifolius* and *L. albus*. Homologies in sequences between other lupin proteins and allergens from peanut, e.g., conglutin α and peanut allergen Ara h 3 or δ-conglutins and Ara h 2, indicate their allergenic potential as well. However, immunological studies did not fully support the allergenicity of all conglutin isoforms, most probably due to interindividual differences in IgE binding of lupin proteins, differences in exposure, and impacts caused by food processing. However, recent findings showed evidence that all conglutin subunits are potential allergens [18], which was confirmed in respect to γ-conglutin by the novel research of Aguilera-Insunza et al., 2023 [20]. Data supporting the allergenicity of minor proteins in lupin seeds are still scarce [18].

According to EU Regulation No. 1169/2011, lupin and products thereof have to be declared on commercial food products [21]. Thus, enzyme-linked immunosorbent assays (ELISAs) [22,23,24,25,26,27] and real-time polymerase chain reaction (PCR) assays [28,29,30] have been developed for verifying compliance with legal regulations. In the last decade, LC-MS/MS has gained in importance in food allergen analysis due to its ability to screen for multiple allergens in one and the same run [31,32,33].

Lupin proteins have been reported to be rather stable during food processing [15,34,35]. However, baking, mild oven cooking, and autoclaving were found to reduce the immunoreactivity of lupin proteins [36]. To date, studies investigating the detectability of lupin in heat-treated samples were carried out either on lupin protein isolates [37], model foods incurred with lupin [24,25,26,29,30,31,32,33,38], or commercial food products [22,23,28]. In the present study, we aimed to investigate the influence of roasting pure lupin seeds on the detectability of lupin proteins by sodium dodecyl sulfate-polyacrylamide gel electrophoresis (SDS-PAGE), ELISAs, and LC-MS/MS, and the detectability of lupin DNA by real-time PCR. Pure roasted lupin seeds play an increasing role in the production of coffee substitutes. The study was performed on seeds of the *L. angustifolius* cultivar “Boregine”, since roasted seeds of this cultivar are used for “lupin coffee” production in a small-sized company in the south of Austria. Roasting was carried out in a professional fluidized bed roasting system. To follow the thermally induced changes of the lupin seeds, samples were drawn during the roasting process at eight temperature levels ranging from 98 °C to 242 °C.

## 2. Materials and Methods

### 2.1. Samples

*L. angustifolius* L., cultivar “Boregine” was grown in the south of Austria and harvested in July 2022. Seeds were designated for use as coffee substitutes. Seeds were roasted in a professional pilot coffee roasting machine (Novaroast BS2, Novaroast GmbH, Lübeck, Germany) and exclusively used for the production of a “lupin coffee” substitute. Roasting was performed in batches of about 2 kg under strict temperature control by following both the temperature of the air stream and the surface temperature of the seeds. Samples were drawn when the seeds reached a surface temperature of 98 °C, 120 °C, 140 °C, 160 °C, 175 °C, 195 °C, 220 °C, and 242 °C, respectively. For the production of the commercially available product, the roasting process is stopped when a seed surface temperature of 220 °C is reached. For this study, we prolonged the roasting process to a final surface temperature of 242 °C to follow potential reactions in the lupin seeds upon high thermal load. Raw (unroasted) lupin seeds of the same batch were used as reference. After roasting, the samples were let to cool down and stored in glass containers at room temperature in the dark until further use.

### 2.2. Protein Analysis

#### 2.2.1. Sample Preparation for SDS-PAGE and ELISAs

First, 10.0 g aliquots of raw and roasted lupin seeds were ground in an electric coffee mill (Model TSM6A013B, Bosch, Giengen, Germany) for 30 s to 60 s. Then, 300.0 mg aliquots of homogenized seeds were weighed out in six 2 mL Eppendorf tubes, followed by adding 1.5 mL of extraction buffer (0.1 M Tris/0.5 M glycine, pH 8.7) to each of the tubes. After vortexing, the Eppendorf tubes were placed into a thermoshaker (Thermomixer C, Eppendorf, Hamburg, Germany), and the mixtures were incubated at 45 °C and 600 rpm for 3 h by vortexing the tubes every 30 min. After centrifugation (Model 5424, Eppendorf) at 10,000 rpm for 15 min, the supernatants were removed and combined in a 15 mL Falcon tube. The protein concentration of the extracts was determined by the Bradford assay using bovine serum albumin (BSA) as a standard. The extracts were stored at 4 °C until further use.

#### 2.2.2. SDS-PAGE

Proteins were denatured at 95 °C for 5 min and separated in 10% polyacrylamide gels using a Mighty Small SE 250 apparatus (Hoefer Scientific Instruments, Holliston, MA, USA). Protein solutions (1 mg/mL) containing BSA (66 kDa), ovalbumin (45 kDa), human IgG (heavy chain 50 kDa, low chain: 25 kDa), or myoglobin (17 kDa) were used as molecular weight markers. Electrophoresis was performed at 40 mA for ~1.5 h. Gels were stained with colloidal Coomassie Brilliant Blue R-250.

#### 2.2.3. ELISAs

The competitive ELISA (IgY ELISA) [25] and the sandwich ELISA [26] were developed in house. Protein extracts from raw and roasted seeds were serially diluted to concentrations ranging from 0.1 ng/mL to 1.0 mg/mL and analyzed in triplicates.

For evaluating the detectability of proteins in roasted lupin seeds, absorbance (A) was normalized as follows:

Competitive ELISA:$$Normalized\;absorbance\;\left[\%\right]=\frac{A-NSB}{A_{0}-NSB}\;100$$

NSB (non-specific binding): absorbance obtained for protein extract from raw seeds, diluted to 1.0 mg/mL

A_0_: absorbance obtained for phosphate-buffered saline (PBS)



Sandwich ELISA:$$Normalized\;absorbance\;\left[\%\right]=\frac{A-NSB}{A_{max}-NSB}\;100$$

NSB: absorbance obtained for PBS

A_max_: absorbance obtained for protein extract from raw seeds, diluted to 1.0 mg/mL



Non-linear regression was carried out using a sigmoid four parametric logistic function by using OriginPro 2020. The detectability of protein from roasted seeds was calculated by dividing the concentration of raw lupin seeds at 50% normalized absorbance, c_raw(50%)_, by the concentration of roasted lupin seeds at 50% normalized absorbance, c_roasted(50%)_, and multiplying by 100:$$Detectability\;\left[\%\right]=\frac{c_{raw\;(50\%)}}{c_{roasted\;(50\%)}}100$$

#### 2.2.4. Sample Preparation for LC-MS/MS

Aliquots of raw and roasted Boregine seeds were homogenized in a laboratory scale blender (BÜCHI mixer B400, BÜCHI Labortechnik AG 9230, Flawil, Switzerland). Samples were further homogenized by milling (IKT 10A, basic version). The total milling time was 2 min, with a 2 min break every 30 s to avoid heat impact on samples.

Then, 0.5 g was extracted in triplicate with a 25 mL buffer (2 N urea, 0.2 N Tris-HCl, pH 9.2) by overhead shaking for 30 min, followed by sonication for 15 min at room temperature [39]. After centrifugation (Heraeus Multifuge 1S, Thermo Fisher Scientific, Munich, Germany) for 10 min at 4000 rpm, supernatants were filtered (0.45 µm regenerated cellulose). Soluble protein was quantified by the Bradford assay at 595 nm using a Rotiquant solution (Carl Roth, Karlsruhe, Germany) containing Coomassie Blue dye. Then, 50 µL of diluted sample extracts (10 to 20 fold) were mixed with 200 µL Rotiquant solution according to the instructions of the manufacturer for the microtiter assay. Calibration was performed with BSA in the range from 5 to 100 µg/mL. All measurements were performed in triplicate for each extract.

Reduction, alkylation, and tryptic digestion were performed according to Geisslitz et al. with some modifications [40]. Due to the higher protein content of lupin compared to wheat, a lower extract volume of 30 µL was used. Furthermore, digestion and SPE purification were adjusted for higher protein and peptide concentrations. In detail, the concentration of dithioerythrol (DTE) was increased to 60 mM and iodoacetamide to 120 mM. Stage-Tips were loaded with six layers instead of five layers, as used previously.

#### 2.2.5. LC-MS/MS

Separation and data acquisition were performed according to Geisslitz et al. [40]. SONAR mode with a 35 Da window was applied for data-independent acquisition (DIA). All samples were measured in triplicate. First, the raw data were processed in ProteinLynx Global server (PLGS) software (version 2.3) from WATERS via the Apex 3D algorithm for deconvolution and lock mass correction, followed by databank search queries with a fasta file containing 110 reviewed entries of lupine proteins downloaded on 9 December 2022. Label free quantification (LFQ) was performed based on exact mass retention time (EMRT) signatures generated for every eluting signal detected, followed by the expression analyses workflow of PLGS (raw lupin was assigned as “normal” and roasted samples as “modified”). The generated protein tables were manually checked for unique peptides based on the used fasta file for lupin. Peptides containing missed cleavage were removed, and normalized intensities obtained from the expression workflow of PLGS were used to calculate relative intensities based on the untreated/raw lupin seeds, which was set as 1.0. The final calculations and heat maps were prepared in excel. A detailed setup of separation, data acquisition, and processing are listed in the Appendix A.

### 2.3. DNA Analysis

#### 2.3.1. Sample Preparation for Real-Time PCR

DNA was extracted from aliquots of raw and roasted lupin seeds that were homogenized with the electric coffee mill. For DNA extraction, the NucleoSpin Plant II DNA kit (Macherey-Nagel, Düren, Germany) was used. A total of 150 mg aliquots of homogenized lupin seeds were weighed out, 900 µL of PL2 buffer was added, and DNA was isolated according to the protocol given by the manufacturer. Isolated DNA was quantified using either the Qubit dsDNA HS Assay Kit or the Qubit dsDNA BR Assay Kit with the Qubit 4 instrument (Thermo Fisher Scientific).

#### 2.3.2. Real-Time PCR Assay for Lupin

Primer and probe sequences were taken from the study by Demmel et al. [28]. Real-time PCR was performed with the QuantiNOVA PCR Kit (Qiagen, Hilden, Germany) on the Rotor-GeneQ thermocycler (Qiagen). The total volume was 20 µL, containing 2 µL of DNA extract (2.5 ng/µL). DNA extracts with lower DNA concentrations were used undiluted. The concentration of forward primer, reverse primer, and the probe was 375 nM. The temperature program was as follows: 2 min at 95 °C (initial denaturation), 45 cycles of 5 s at 95 °C (denaturation), and 5 s at 60 °C (combined annealing and elongation).

#### 2.3.3. Statistical Analysis

One-way analysis of variance (ANOVA) followed by a posthoc test according to Scheffe was applied at the p level of 0.05 using SPSS software (IBM, Version 26).

## 3. Results

With increasing roasting temperature, the color of Boregine seeds changed from beige to dark brown, as expected (Figure 1). The distinct browning of seeds started at a temperature of 195 °C; seeds roasted at 242 °C were almost black.

### 3.1. Protein Analysis

#### 3.1.1. Protein Extraction

Two protocols were used for extracting proteins, extraction with 0.1 M Tris, 0.5 M glycine, and pH 8.7 buffer for 3 h (extraction protocol 1, slightly modified from [26]) and extraction with 2 N urea, 0.2 N Tris-HCl, and pH 9.2 for 30 min, followed by sonication for 15 min (extraction protocol 2). Extraction with Tris-glycine buffer resulted in lower protein recoveries for roasted seeds compared to raw seeds (Figure 2a). Recoveries for seeds roasted at 98 °C, 120 °C, 140 °C, 160 °C, and 175 °C were 77.2%, 70.4%, 50.0%, 37.3%, and 17.3%, respectively. For seeds roasted at temperatures ≥195 °C, recoveries were ≤5.1%.

In general, recoveries obtained with extraction protocol 2, involving 2 N urea and a sonication step, were higher than those obtained with extraction protocol 1 (Figure 2b). The highest recovery (124.4%) was obtained for seeds roasted at 98 °C. Seeds roasted at 120 °C, 140 °C, 160 °C, 175 °C, and 195 °C resulted in recoveries of 84.8%, 85.0%, 47.9%, 32.4%, and 19.7%, respectively. For seeds roasted at temperatures ≥220 °C, recoveries were ≤8.2%.

#### 3.1.2. SDS-PAGE

Protein extracts obtained with extraction protocol 1 were subjected to SDS-PAGE. Extracts from seeds roasted at temperatures ≤175 °C resulted in profiles consisting of multiple bands (Figure 3a,b). Seeds roasted at 195 °C led to one single band at the dye front, whereas for extracts from seeds roasted at temperatures ≥220 °C, not any bands were obtained (Figure 3b).

Protein profiles for roasted seeds differed from the profile obtained for raw seeds. Bands at ~85 kDa, ~68 kDa, and ~59 kDa were obtained up to roasting temperatures of 98 °C, 160 °C, and 175 °C, respectively (Figure 3a). A band at ~51 kDa was most pronounced after roasting at 98 °C. Bands at ~48 kDa and ~38 kDa were obtained up to a roasting temperature of 175 °C and a band at ~17 kDa up to 195 °C. A strong band at ~28 kDa was observed up to a temperature of 120 °C and disappeared in extracts from seeds roasted at higher temperatures.

#### 3.1.3. ELISAs

Protein extracts obtained with extraction protocol 1 were subjected to a competitive and a sandwich lupin ELISA. With both ELISAs, lupin proteins could be detected even in seeds roasted at 242 °C; however, the detectability for proteins from seeds roasted at temperatures ≥195 °C was drastically lower compared to that for proteins from raw seeds and seeds roasted at lower temperatures (Figure 4). For the competitive ELISA, detectabilities of 141.8%, 145.1%, 114,2%, 81.4%, 205.4%, 0.8%, 1.3%, and 0.7% were determined for roasting temperatures of 98 °C, 120 °C, 140 °C, 160 °C, 175 °C, 195 °C, 220 °C, 242 °C, respectively (Figure 4a). The sandwich ELISA showed detectabilities of 181.9%, 311.9%, 202.4%, 95.0%, 37.2%, 0.4%, 0.9%, and 3.2%, respectively (Figure 4b).

#### 3.1.4. LC-MS/MS

For the identification of proteins via their tryptic peptides, only oxidation of methionine was applied as a variable modification, which is frequently used in untargeted proteomics, to avoid distortion of the results. Results of LFQ are presented as heat maps for lupin protein groups and their isoforms (Figure 5). All major lupin seed proteins, belonging to a group of conglutins, and four other proteins were detected with sufficient scores and intensities for accurate LFQ. Based on the used approach, single isoforms of alpha, beta, gamma, and omega conglutins can be distinguished. Minor allergens such as PR-10 proteins (Lup a 4 and Lup l 4) were not detected; nsLTP (Lup an 3) and profilins (Lup a 5) were not considered because they are not included as reviewed proteins in Uniprot [18].

All α-, γ-, and β-conglutins, with the exception of the β 1 protein, were quantified with a high number of peptides. δ-conglutins have highly overlapping sequences, which delivered only two to three unique selective peptides for the single isoforms. Conglutin δ 1 and 3 were not even discriminable and thus quantified together with two shared peptides and one unique peptide, SSQESEESEELDQCCEQLNELNSQR, for conglutin δ 1.

The group of α-conglutins was quantified with similar abundances compared to the control up to a roasting temperature of 140 °C to 175 °C, depending on the single isoform. Conglutin α 3 revealed the highest thermal stability and was measurable up to a roasting temperature of 220 °C. β-conglutins showed quite a high temperature stability up to 175 °C with the exception of isoform 1, for which only one peptide was assigned for quantification. For isoform conglutin β 5 and 7, even the highest abundances were detected at 175 °C. The extraordinary high relative abundance of conglutin β 5 at 175 °C was predominantly caused by two major peptides; intensities of LLGFGINADENQR was doubled and NFLAGSEDNVIR was one and a half times higher compared to the untreated control. The effect of roasting among δ-conglutins was heterogeneous. Conglutin δ 1 and 3 showed superior abundances from 140 °C to 175 °C, whereas the conglutins δ 2 and 4 delivered strongly reduced intensities at 160 °C and 175 °C, respectively. Contrary to other conglutins, the group of γ-conglutins disposed instability towards heat exposure. Already at 120 °C, moderate and at 140 °C, intense reductions of relative abundance were notable. The other proteins not belonging to the allergenic conglutins had quite low thermal stabilities, which were already visible in a moderate to strong decrease at 120 °C. The slightly higher abundance at 98 °C roasting for P69590 and O49884 can be explained by the increased extraction rates of proteins at this temperature as determined by Bradford assay. In addition, the significantly lower concentrations of these proteins compared to the conglutins should be taken into account. The large difference in concentration may yield biased results in terms of thermal stability.

### 3.2. DNA Analysis

#### 3.2.1. DNA Extraction

The influence of roasting temperature of lupin seeds on DNA extractability is shown in Figure 6a. For seeds roasted at 98 °C, a higher DNA yield (118.5%) was obtained than for raw seeds. Roasting temperatures of 120 °C and 140 °C resulted in DNA recoveries of 83.5% and 68.0%, respectively. A drastic drop in DNA yield was observed at a roasting temperature of 160 °C, with DNA recoveries for seeds roasted at 160 °C, 175 °C, and 195 °C being 16.2%, 20.0%, and 7.9%, respectively. DNA concentrations of extracts from seeds roasted at ≥220 °C were below the LOQ of the Qubit dsDNA HS Assay Kit.

#### 3.2.2. Real-Time PCR

The influence of roasting on the detectability by real-time PCR is shown in Figure 6b. An increase in the fluorescence signal was observed for all seed samples, even for those roasted at higher temperatures. However, DNA extracts from seeds roasted at temperatures ≥ 175 °C resulted in higher Ct values (∆ Ct ~5) compared to raw seeds and seeds roasted at temperatures ≤ 160 °C.

## 4. Discussion

Due to the small size and round shape of Boregine seeds, they could be roasted more homogeneously in the fluidized bed roasting system than other lupin species and cultivars. The seeds were roasted until surface temperatures of 98 °C, 120 °C, 140 °C, 160 °C, 175 °C, 195 °C, 220 °C, and 242 °C, respectively, were reached. With increasing roasting temperature, the color changed from beige to dark brown/black. Distinct browning was observed for seeds with surface temperatures ≥ 195 °C. Browning occurred due to the Maillard reaction, a cascade of complex interactions between reducing sugars and amino groups of amino acids, peptides, and proteins due to thermal processing, resulting in a complex mixture of Maillard reaction products [41]. The primary reaction of the initial stage is the condensation between the carbonyl groups of reducing carbohydrates and amino groups within proteins, forming glycosylated proteins. Predominantly, the ε-amino group of lysine, and, to a lesser extent, the α-amino groups of terminal amino acids react. Furthermore, imidazole and indole groups of histidine and tryptophan can be involved to a much lower degree [42]. In the late phase of the Maillard reaction, heterogeneous brown pigments, so-called melanoidins, are formed by cyclization, dehydration, rearrangement, isomerization, and condensation of low molecular weight Maillard reaction products [43]. Melanoidin formation strongly depends on the type of food, the composition of amino acids and reducing sugars, heating temperature, and heating time [41].

Two protocols were applied for extracting lupin proteins from raw and roasted seeds. Protocol 1 was based on extracting proteins with 0.1 M Tris, 0.5 M glycine, pH 8.7 buffer at 45 °C for 3 h. Protocol 2 encompassed incubation with 2 N urea, 0.2 N Tris-HCl, pH 9.2 at room temperature for 30 min, followed by sonication for 15 min. Alkaline buffers are frequently used for extracting proteins from legume seeds since they result in higher protein yields than neutral or acidic buffers [44]. With the urea-based extraction buffer (protocol 2), higher protein yields were obtained than with the buffer lacking urea (protocol 1), which can be explained by chaotropic properties of urea for solubilizing protein and is in line with the literature [23]. Additionally, the ultrasonic treatment enhanced solubility due to disulfide bond breakage [45]. However, since urea is a denaturing agent, we had to refrain from its usage when extracts were subsequently analyzed by ELISAs. For ELISA analyses, extracts were prepared with a Tris-glycine based buffer. Since Tris-glycine does not show denaturing effects, it is compatible with analyses involving interactions between antigens and antibodies [23,25,26]. Both extraction protocols reflect very well the individual circumstances/conditions of applied methodologies for protein quantification. On the one hand, mild conditions have to be applied for ELISA analyses to preserve protein structures; on the other side, for protein analysis by LC-MS/MS, extraction under reducing and denaturing conditions is used, commonly applied before digestion in order to improve breakdown of proteins to peptides as well.

Roasting had a strong impact on the yield of lupin proteins. In the case of extraction protocol 1, protein yields gradually decreased with increasing roasting temperature. By applying extraction protocol 2, the highest yield was obtained for seeds roasted at 98 °C, indicating that slight heat treatment increased the solubility of certain lupin proteins. In general, free but also protein-bound amino acids are important reaction partners during thermal treatment of foods, particularly in the final stages of the Maillard reaction [41]. The decrease in amino acids during thermal treatment is dependent on the thermal load. Celik and Gökmen, 2020 described a decrease in the concentration of free amino acids in bread depending on the baking time [46]. Casal et al., 2005 observed a decrease in free amino acids during the roasting of Arabica coffee, with the largest decrease in concentrations in the temperature range between 160 °C and 180 °C [47]. Heat treatment may also lead to protein aggregation, resulting in decreased protein solubility.

Subjecting lupin protein extracts obtained with extraction protocol 1 to SDS-PAGE let to protein profiles containing multiple bands. For raw lupin seeds, bands at ~85 kDa, ~68 kDa, ~59 kDa, ~51 kDa, ~48 kDa, ~38 kDa, ~28 kDa, and ~17 kDa were obtained. The molecular weights correspond quite well with those previously reported for lupin conglutins under denaturing conditions. Most probably, bands at ~59 kDa, ~48 kDa, ~28 kDa, and ~17 kDa can be assigned to β-conglutin, α-conglutin, γ-conglutin, and δ-conglutin, respectively, which is in high accordance with the results of Foley et al., 2015 [48]. The non-specific lipid transfer protein was probably not assigned due to its smaller size of 11 kDa. With increasing roasting temperature, the number of bands decreased. Seeds roasted at 195 °C led to only one band at the dye front, whereas for extracts from seeds roasted at temperatures ≥220 °C, not any bands were obtained. These results can be explained by three factors, the loss of amino acids in the Maillard reaction, differences in the thermal stability of amino acids, and the binding preference of Coomassie Brilliant Blue for cationic amino acids. Casal et al. showed that by coffee roasting, the concentration of all amino acids decreased, although at different rates [47]. Tyrosine, valine, leucine, phenylalanine, and alanine turned out to be more stable, whereas lysine, methionine, and histidine were rapidly lost in the browning reaction [47]. By exposing three proteins (lysozyme, ribonuclease, and insulin) to dry heating at temperatures between 80 °C and 180 °C for 1 h to 24 h, Weder et al. found most amino acids to be stable up to 120 °C [49]. Up to ~150 °C, decomposition of amino acids was almost rectilinear with temperature and time, and at 160 °C, a critical temperature was reached for most of the amino acids [49]. Nonpolar aliphatic, acidic, and aromatic amino acids were relatively stable; the lability of the other amino acids increased in the order proline, arginine, histidine, cysteine, threonine, lysine, tryptophan, serine, and methionine [49]. The color intensity of complexes of proteins with Coomassie Brilliant Blue, frequently used for staining proteins on SDS-polyacrylamide gels, strongly depends on the number of lysine, histidine, and arginine residues [50], all three being relatively thermolabile and rapidly lost due to the Maillard reaction.

The two ELISAs used in this study were developed in-house [25,26]. Although IgY and IgG antibodies were raised against a protein extract from *L. albus*, the ELISAs showed cross-reactivity with *L. angustifolius* cultivars [25,26]. The competitive ELISA showed 45% detectability for (raw) seeds from the cultivar Boregine compared to (raw) seeds from *L. albus* [25]. With both ELISAs, lupin proteins could be detected in all roasted seed samples. However, both ELISAs showed drastically lower detectability for proteins from seeds roasted at temperatures ≥ 195 °C compared to proteins from raw seeds and seeds roasted at lower temperatures. The applicability of the ELISAs to detect proteins from *L. albus* in heat-treated foods was demonstrated previously by analyzing model foods, including bread, biscuits, and pasta [25,26].

Protein extracts obtained with extraction protocol 2 were subjected to LC-MS/MS. The found and selected peptides were mainly in accordance with other studies [31,51]. The mentioned circumstances and difficulties to find unique peptides for δ-conglutins were already described by Tahmasian et al. [51]. Similar outcomes were presented by a study of Downs et al., 2016, who used LFQ to examine heat impact and extraction efficiency for detection of allergens from roasted walnuts [52]. Some walnut proteins such as nsLTP (Jug r3) showed reduced abundance after high temperature treatment at 180 °C. On the other side, strongest heat exposure resulted in a markable increased abundance of 7S and 11S globulins. Furthermore, this study included different extraction protocols, which revealed differences among obtained intensities of the analyzed proteins [52]. The influence of processing of food products and the impact of heating on milk, egg, and peanut allergens was studied by Parker et al., 2015 by means of LC-MS/MS and ELISA [33]. Two different types of food, a cereal bar and muffin, were produced, which revealed different peptide intensities among examined allergens depending on the manufacturing process and matrix [33]. Intense effects of processing and heat on nut allergens in food products (bread and cookies) were also reported by Korte et al., 2019 using targeted MRM experiments with a QTRAP instrument. Some peptides were not significantly affected, whereas others showed a strong reduction in recovery rates [53].

In general, the higher the number of peptides and, thus, sequence coverage of the protein, the more reliable results can be obtained. The low number of assigned peptides for conglutin β 1 and the group of δ-conglutins might be responsible for the lower accuracy of the results. Denaturation of proteins affects their structure, solubility, and accessibility for digestion. In respect to proteomics workflow, denaturation should have a minor influence because detergents are used for improving denaturation (in this study, the surfactant Rapigest was used). Nevertheless, denaturation and modifications by heat have a strong impact on solubility, which was shown by the Bradford assay. Furthermore, the Maillard reaction contributes to protein modifications and finally to degradation. The stability of single amino acids has to be considered as well. Methionine, as already mentioned, is the amino acid most susceptible to oxidation. This kind of variable modification is commonly taken into account for untargeted proteomics by applied data processing workflow. The glycosylation of proteins during the initial stage of the Maillard reaction was not considered by the used workflow. However, these circumstances might have minor effects since arginine is barely susceptible for glycosylation via the Maillard reaction and much more abundant than lysine in lupine seeds. At around 160 °C, first amino acids start to decompose, whereas most nonpolar aliphatic, acidic, and aromatic amino acids are all relatively stable until the mentioned thermal load [49,54]. The observed detectability of conglutin protein fractions reflects very well the individual thermal stability of amino acids. β-conglutins revealed the highest thermal stability due to a high share of leucine and phenylalanine; concurrently, sulfur-containing amino acids (methionine and cysteine) are found only in traces. In α-conglutins, leucine and isoleucine are the dominant amino acids, whereas methionine is found in low concentrations (around 0.2%). γ- and δ-conglutins displayed thermally labile behavior due to high shares of threonine and cysteine, respectively. Furthermore, both fractions are rich in methionine (around 0.5–0.8%) [17]. This conclusion was further supported by the detectability of single peptides such as LLGFGINADENQR originating from conglutin β 5. This peptide showed the highest thermal stability and abundance at 178 °C, probably due to the absence of labile amino acids.

A comparison of ELISA and LC-MS/MS data reveals that in contrast to LC-MS/MS, both ELISAs allowed detecting lupin proteins, even in lupin seeds roasted at 242 °C, although to a very low extent. Both IgG and IgY were obtained by immunization with a total protein extract of (white) lupin seeds. Most probably, a proportion of the antibodies was raised against proteins with rather stable epitopes.

To directly assess the allergenic potential based on the peptide abundance of single proteins, it is essential to connect peptide and epitope sequences. To date, only a small number of studies assigned epitope sequences of allergens from lupin seeds. Lima-Cabello et al., 2016 applied in silico tools for allergen structure analysis to identify T- and B-cells epitopes due to their conformational IgE-binding. Intensive modeling predicted four epitopes within β-conglutins, NFRLLGFGIN (IgE1), KGLTFPGSTE (IgE2), RRYSARLSEG (IgE3), and SYFSGFSRNT (IgE4) [55]. A major drawback lies in the fact that the enzyme used for cleavage generates differing peptides (trypsin cleavages proteins at R and K). Nevertheless, IgE1 was partially recognized in LLGFGINANENQR of β-conglutins 7 and IgE4 in DQQSYFSGFSK of β-conglutins 2, 3, 4, and 6. More administrable were results obtained for suitable target antigens after immunocapture and LC-MS/MS analysis of Lup an 1, as shown by Hu et al., 2021 [56]. Two peptides were in accordance with this study, LLGFGINADENQR and DQQSYFSGFSK. Both peptides were detectable and stable until 175 °C. These outcomes and implications indicated that the mentioned peptides were suitable as selective markers to estimate the immunogenic potential of lupins in processed food based on the major allergens of β-conglutins.

In order to investigate the detectability of lupin DNA from roasted Boregine seeds by real-time PCR, DNA was extracted using a commercial DNA extraction kit for plants. For seeds roasted at 98 °C, a higher DNA yield was obtained than for raw seeds, which is in line with the higher protein yield obtained with protein extraction protocol 2. For seeds roasted at higher temperatures, lower DNA yields were obtained. The DNA concentration of extracts from seeds roasted at ≥220 °C were below the LOQ.

In general, DNA is more stable than proteins. However, food processing, including heat treatment, may result in fragmentation of DNA and thus lower amplifiability by PCR. The primers designed for *L. angustifolius* by Demmel et al. [28] result in a rather short amplicon of 129 bp. The applicability of the real-time PCR assay for processed foods was investigated by a variety of commercial food products, including various bakery products, fruit-flavored gums, and ice cream [28] and model pizza [38]. Roasting of Boregine seeds had an impact on the detectability by real-time PCR. For all roasted seed samples, an increase in the fluorescence signal was observed. However, for seeds roasted at temperatures ≥ 175 °C, higher Ct values were obtained (∆ Ct ~5) than for raw seeds and seeds roasted at lower temperatures.

## 5. Conclusions

Thermal treatment of lupin seeds as performed for the production of “lupin coffee” showed significant effects on the extractability and detectability of lupin proteins and DNA, starting with a roasting temperature of 140 °C. By LC-MS/MS, β-conglutin could not be detected in lupin seeds that were roasted at temperatures higher than 195 °C, which is a thermal load that is necessary to obtain the desired roasted flavor of the product. This is of special interest, as the allergenicity of *L. angustifolius* is predominantly assigned to β-, and to a lower extent to α-conglutin reactions [19]. β-conglutin isoforms have been reported as potential allergenic triggers, whereas other conglutin families are unlikely to elicit allergic reactions [48].

Our study focused on investigating the impact of heat treatment of pure lupin seeds on the detectability of lupin proteins and lupin DNA. The transferability of our findings to other heat-treated lupin food products including pasta, bread, and cookies is limited because the presence of further matrix compounds may have an impact on extractability and detectability of both proteins and DNA.

Our results suggest reduced allergenicity of roasted lupin seeds used for the production of “lupin coffee”, which might make “lupin-coffee” suitable for consumers suffering from an allergy towards lupin allergens. However, the effect of thermal processing to reduce the allergenic potential of lupin proteins is not clear, and in the literature, ambiguous outcomes were presented, according to the review of Villa et al., 2020 [18]. Furthermore, minor lupin allergens were not detected by LC-MS/MS and most probably also not by the ELISAs. Nevertheless, the applied methodologies are implemented in routine analysis, and the combination of them improved the validity of the results. LC-MS/MS can achieve a similar level of informative value if the corresponding marker peptides have high sequence similarities to epitopes. In addition, untargeted methods offer the advantage of detecting all proteins, which means that even yet unknown allergens can be included on the basis of suitable peptides.

It is noteworthy that all presented results refer to the (roasted) lupin seeds as such and not to the final coffee brew. The results on protein extractability suggest even lower concentrations in the final brew than in the roasted seeds. However, it has to be emphasized that the results that were obtained in this investigation can only estimate the reduction in allergenicity. For verification, in vivo tests are required to confirm the obtained results.

## Figures and Tables

**Figure 1 foods-13-00673-f001:**
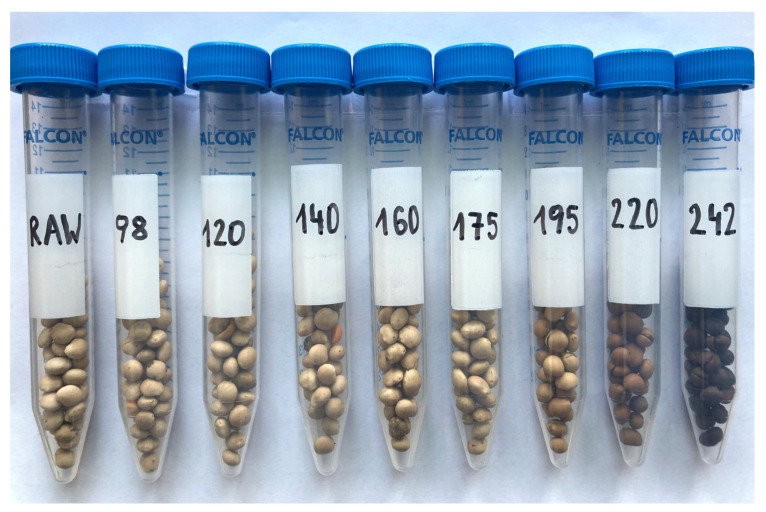
Seeds of lupin (*L. angustifolius* L., cultivar “Boregine”) before and after roasting in a professional pilot coffee roasting machine. Samples were drawn at surface temperatures of 98 °C, 120 °C, 140 °C, 160 °C, 175 °C, 195 °C, 220 °C, and 242 °C, respectively.

**Figure 2 foods-13-00673-f002:**
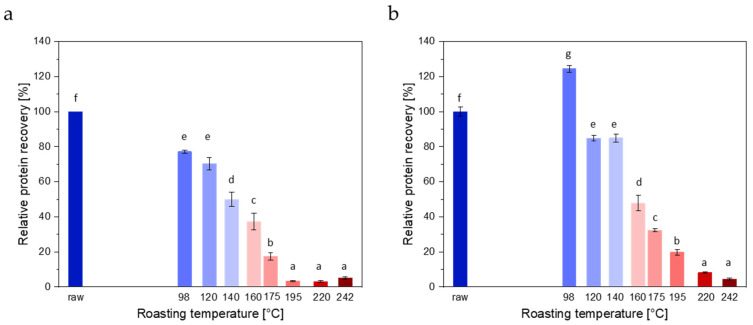
Influence of roasting temperature on protein recovery. (**a**) Extraction protocol 1: 0.1 M Tris, 0.5 M glycine, pH 8.7; 3 h. Mean and standard deviation of four replicates. (**b**) Extraction protocol 2: 2 N urea, 0.2 N Tris-HCl, pH 9.2; 30 min, followed by sonication for 15 min. Mean and standard deviation of three replicates. Small letters indicate homogeneous subgroups based on ANOVA (*p* ≤ 0.05) and post hoc test according to Scheffe.

**Figure 3 foods-13-00673-f003:**
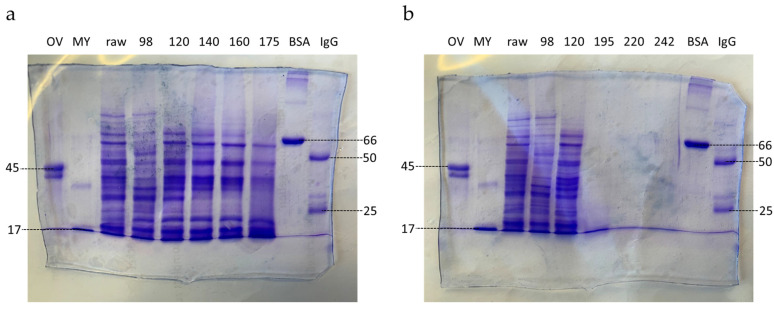
SDS-PAGE of extracts obtained with extraction protocol 1 from raw and roasted lupin seeds. (**a**): seeds roasted at 98 °C, 120 °C, 140 °C, 160 °C, 175 °C; (**b**): seeds roasted at 98 °C, 120 °C, 195 °C, 220 °C, 242 °C. Extracts from raw seeds and seeds roasted at temperatures ≤ 98 °C were diluted to a protein concentration of 8.3 mg/mL; extracts from seeds roasted at higher temperatures were loaded undiluted. OV: ovalbumin (45 kDa), MY: myoglobin (17 kDa), BSA: bovine serum albumin (66 kDa), IgG: human immunoglobulin G (heavy chain: 50 kDa, low chain: 25 kDa). Protein sizes of the standard proteins (in kDa) are indicated on the left and right side of the gels.

**Figure 4 foods-13-00673-f004:**
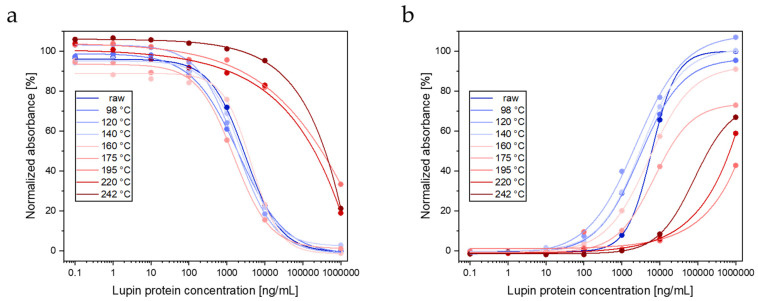
Influence of roasting temperature on the detectability by lupin ELISAs. (**a**) competitive ELISA, (**b**) sandwich ELISA. Protein extracts from raw and roasted lupin seeds were obtained with extraction protocol 1.

**Figure 5 foods-13-00673-f005:**
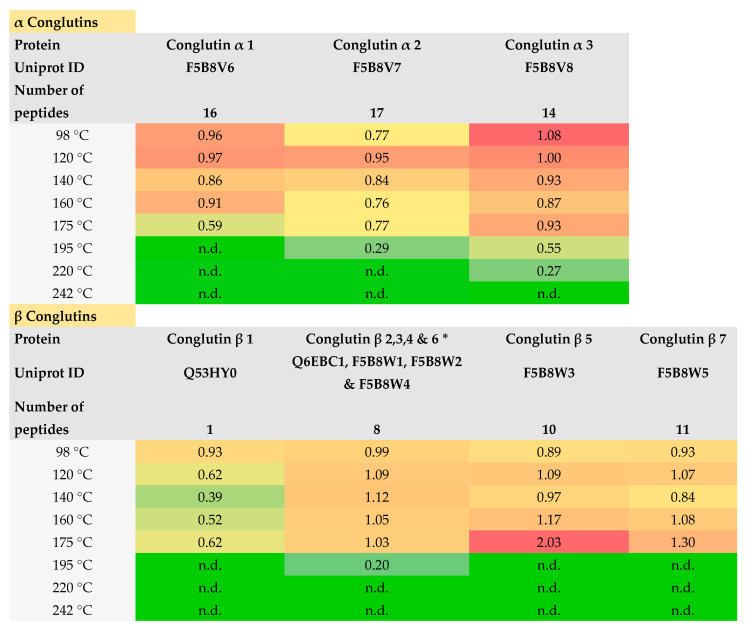
Heatmap showing the influence of roasting temperature on relative intensities based on raw lupine via LFQ by LC-MS/MS. Warmer tones like red and orange indicate higher values, while cooler tones like yellow and green represent lower values. * based on shared peptides, no unique peptides found; n.d. not detected.

**Figure 6 foods-13-00673-f006:**
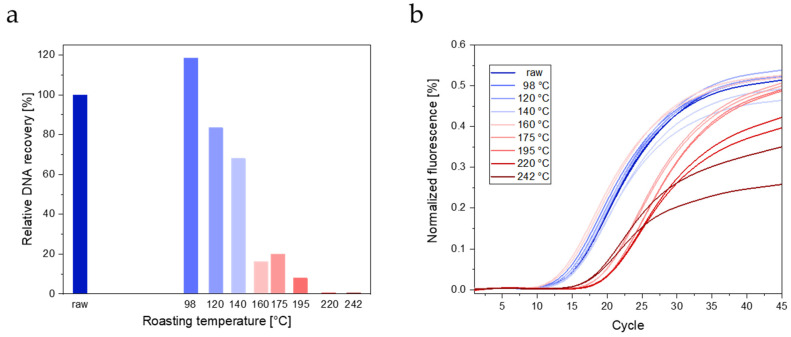
Influence of roasting temperature on (**a**) DNA recovery (means of two technical replicates) and (**b**) amplification by real-time PCR.

## Data Availability

The via PLGS processed data from untargeted LC-MS/MS measurements are exported as mzid files and stored in the Appendix A. Raw data files will be made available on request.

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
