# Peer review of "Influence of Roasting Temperature on the Detectability of Potentially Allergenic Lupin by SDS-PAGE, ELISAs, LC-MS/MS, and Real-Time PCR"

_foods, 2024, doi:10.3390/foods13050673_

Round 1

Reviewer 1 Report

Comments and Suggestions for Authors

Beyer et al. investigated the association between roasting and presence of protein in lupin. Although lupin is allergic to sensitive individuals, this manuscript only showed the change of total protein in lupin. As authors mentioned Lup an 1, Lup an 3 and Lup an 5 are allergens in lupin, no experiments and results related to allergens were discussed throughout the article. The study did not elucidate the influence of IgE binding ability and allergen content after roasting. It is hard to connect the relationship between roasting and allergenicity of lupin. Other queries are list as below,

1.     The authors should write more information of allergens in introduction.

2.     Data only showed the content of total protein and DNA, not allergen.

3.     What did LC/MS/MS used for? 

4.  Only protein profile was observed by SDS-PAGE. Which one is allergen?

Author Response

We revised the manuscript according to the comments as follows:

  • Beyer et al. investigated the association between roasting and presence of protein in lupin. Although lupin is allergic to sensitive individuals, this manuscript only showed the change of total protein in lupin.

Actually, we investigated the impact of roasting on the extractability and detectability of lupin proteins, not the “presence of protein in lupin”. Furthermore, we did not “only show the change of total protein in lupin”. Both gel electrophoresis and LC-MS/MS yielded information on individual lupin proteins.

  • As authors mentioned Lup an 1, Lup an 3 and Lup an 5 are allergens in lupin, no experiments and results related to allergens were discussed throughout the article. The study did not elucidate the influence of IgE binding ability and allergen content after roasting. It is hard to connect the relationship between roasting and allergenicity of lupin.

We do not understand what the reviewer means with “… no experiments and results related to allergens were discussed throughout the article”, because LC-MS/MS data provided information on the impact of heat treatment on individual lupin allergens. However, we agree with the reviewer that the original version of the manuscript did not give any information on the relationship between heat treatment and allergenicity. We therefore added a paragraph focusing on this issue.

To directly assess the allergenic potential based on peptide abundance of single proteins it is essential to connect peptide and epitope sequences. To date, only a small number of studies assigned epitope sequences of allergens from lupin seeds. Li-ma-Cabello et al., 2016 applied in silico tools for allergen structure analysis to identify T- and B-cells epitopes due to their conformational IgE-binding. Intensive modeling predicted four epitopes within β-conglutions, NFRLLGFGIN (IgE1), KGLTFPGSTE (IgE2), RRYSARLSEG (IgE3), and SYFSGFSRNT (IgE4) [53]. Major drawback relies in the fact that the enzyme used for cleavage generates differing peptides (trypsin cleav-ages proteins at R and K). Nevertheless, IgE1 was partially recognized in LLGFGINANENQR of β-conglutins 7 and IgE4 in DQQSYFSGFSK of β-conglutins 2, 3, 4, and 6. More administrable were results obtained for suitable target antigens after immunocapture and LC-MS/MS analysis of Lup an 1 as shown by Hu et al., 2021 [54]. Two peptides were in accordance with this study,  LLGFGINADENQR and DQQSYFSGFSK. Both peptides were detectable and stable until 175 °C. These out-comes and implications indicated that the mentioned peptides were suitable as selec-tive markers to estimate the immunogenic potential of lupins in processed food based on the major allergens of β-conglutins.

  • The authors should write more information of allergens in introduction.

As suggested by the reviewer, we added more information on lupin allergens in the introduction section as follows:

To date, the World Health Organization (WHO) / International Union of Immunological Societies (IUIS) allergen nomenclature subcommittee and the ALLERGOME (http://www.allergome.org/) databases assign different families of lupin proteins as al-lergens, in detail globulins (α-, β-, and γ-conglutin), 2S albumins (δ-conglutin), and a few minor fractions such as nonspecific lipid transfer proteins (nsLTP) (Lup an 3), pathogenesis-related (PR)-10 proteins (Lup a 4 and Lup l 4), and profilins (Lup a 5) [19]. Studies have shown the presence of several important IgE reactive proteins in lupin, of which β-conglutin was identified as the major allergen in L. angustifolius and L. albus. Homologies in sequences between other lupin proteins and allergens from peanut, e.g. conglutin-α and peanut allergen Ara h 3 or δ-conglutins and Ara h 2, indicate their al-lergenic potential as well. However, immunological studies did not fully support the allergenicity of all conglutin isoforms, most probably due to interindividual differences in IgE binding of lupin proteins, differences in exposure, and impacts caused by food processing.  However, recent findings showed evidence that all conglutin subu-nits are potential allergens [18], which was confirmed in respect to γ-conglutin by the novel research of Aguilera-Insunza et al., 2023 [20]. Data supporting the allergenicity of minor proteins in lupin seeds is still scarce [18].

  • Data only showed the content of total protein and DNA, not allergen.

As mentioned above, the study does provide information on individual lupin allergens.

  • What did LC/MS/MS used for? 

Actually, LC-MS/MS was used to obtain information on the impact of heat treatment on specific proteins.

  • Only protein profile was observed by SDS-PAGE. Which one is allergen?

We agree with the reviewer that this information is important. Thus, we added missing information as follows: Most probably, bands at ~59 kDa, ~ 48 kDa, ~ 28 kDa, and ~ 17 kDa can be assigned to β-conglutin, α-conglutin, γ-conglutin, and δ-conglutin, respectively, which is in high accordance with results of Foley et al., 2015 [45]. The non-specific lipid transfer protein, was probably not assigned due to its smaller size of 11 kDa.”  

Reviewer 2 Report

Comments and Suggestions for Authors

The manuscript titled "Influence of Roasting Temperature on the Detectability of Potentially Allergenic Lupin by SDS-PAGE, ELISAs, LC-MS/MS, and Real-time PCR," authored by Bruno Beyer, Dominik Obrist, Philipp Czarda, Katharina Pühringer, Filip Vymyslicky, Barbara Siegmund, Stefano D´Amico, and Margit Cichna-Markl, illustrates a reduction in lupin extractability and detectability in protein extracts following thermal treatments at various temperatures. The manuscript is well-crafted, employing contemporary techniques. However, it is widely recognized that elevated temperatures, such as those used in roasting, significantly impact food proteins, diminishing their extractability.

Key observations regarding the study include:

1.     The authors utilized DNA extraction and quantification by RT-PCR, a method commonly employed to assess lupin contamination in food. At extreme temperatures, DNA degradation occurs, starting at 130°C, with complete degradation above 190°C, as reported by some authors. Given these facts, what led the authors to choose this methodology?

2.     Two different protocols were applied for protein extraction: Protocol 1 (0.5 M Tris, 0.1 M glycine, pH 8.7, 3h) and Protocol 2 (2 M urea, 0.2M Tris-HCl, pH 9.2, 30 min). Despite Protocol 2 demonstrating superior protein extractability (as shown in Figure 2), the authors chose Protocol 1 for SDS-PAGE and ELISA experiments. The decision to avoid Protocol 2 for these experiments may be attributed to urea interfering with protein binding to antibodies, although it can be eliminated before experimentation, for instance, through extensive dialysis. Why did the authors decide to use protocol 1?

3.     The LC-MS/MS technique lacks a detailed explanation of running conditions in the Supplementary section. The concern is raised regarding the appropriateness of relying on unique peptides for acquisition, as high temperatures may induce significant modifications in peptides that may evade detection by mass spectrometry.

4.     Lupins are detectable by ELISA even after intense thermal treatment using polyclonal antiserum and antiserum precipitated with AS, yet they are not detected by LC-MS/MS. The authors are queried about their interpretation of this disparity.

In summary, the manuscript presents insights into the impact of roasting temperature on lupin detectability using various techniques. The raised points seek clarification on methodological choices and interpretation of results to enhance the scientific rigour of the study.

Comments on the Quality of English Language

The manuscript is written in clear and understandable English, necessitating only minor corrections.

Author Response

We thank reviewer 2 for the helpful comments.

  1. The authors utilized DNA extraction and quantification by RT-PCR, a method commonly employed to assess lupin contamination in food. At extreme temperatures, DNA degradation occurs, starting at 130°C, with complete degradation above 190°C, as reported by some authors. Given these facts, what led the authors to choose this methodology?

Actually, real-time PCR plays an increasing role in food allergen analysis. One of the advantages of DNA based methods is that DNA is generally more stable than proteins. However, it is well known that heat treatment will lead to DNA fragmentation.

  1. Two different protocols were applied for protein extraction: Protocol 1 (0.5 M Tris, 0.1 M glycine, pH 8.7, 3h) and Protocol 2 (2 M urea, 0.2M Tris-HCl, pH 9.2, 30 min). Despite Protocol 2 demonstrating superior protein extractability (as shown in Figure 2), the authors chose Protocol 1 for SDS-PAGE and ELISA experiments. The decision to avoid Protocol 2 for these experiments may be attributed to urea interfering with protein binding to antibodies, although it can be eliminated before experimentation, for instance, through extensive dialysis. Why did the authors decide to use protocol 1?

We agree with the reviewer that urea could be removed by extensive dialysis. However, our aim was to use extraction protocols that are commonly used. Tris glycine buffer (without ure)a is commonly used for extracting lupin proteins for subsequent analysis by SDS-PAGE and/or ELISAs.

  1. The LC-MS/MS technique lacks a detailed explanation of running conditions in the Supplementary section. The concern is raised regarding the appropriateness of relying on unique peptides for acquisition, as high temperatures may induce significant modifications in peptides that may evade detection by mass spectrometry.

As suggested by the reviewer, we added information on experimental details of LC-MS/MS,
 as follows:

“Separation and data acquisition was performed according to Geisslitz et al. [39]. SONAR mode with 35 Da window was applied for data independent acquisition (DIA). All samples were measured in triplicate. First, the raw data was processed in ProteinLynx Global server (PLGS) software (version 2.3) from WATERS via the Apex 3D algorithm for deconvolution and lock mass correction, followed by databank search querries with a fasta file containing 110 reviewed entries of lupine proteins downloaded on 9th December 2022. Label free quantification (LFQ) was performed based on exact mass retention time (EMRT) signatures generated for every eluting signal detected, followed by the expression analyses workflow of PLGS (raw lupin was assigned as “normal” and roasted samples as “modified”). The generated protein tables were manually checked for unique peptides based on the used fasta file for lupin. Peptides containing missed cleavage were removed and normalized intensities obtained from the expression workflow of PLGS were used to calculate relative intensities based on the untreated/raw lupin seeds, which was set as 1.0. The final calculations and heat maps were prepared in excel. Detailed setup of separation, data aquisistion and processing are listed in the supplement (see Supplementary File).”

  1. Lupins are detectable by ELISA even after intense thermal treatment using polyclonal antiserum and antiserum precipitated with AS, yet they are not detected by LC-MS/MS. The authors are queried about their interpretation of this disparity.

As proposed by the reviewer, we discussed disparity of ELISA and LC-MS/MS data by adding the following paragraph:

Comparison of ELISA and LC-MS/MS data reveals that in contrast to LC-MS/MS, both ELISAs allowed detecting lupin proteins even in lupin seeds roasted at 242 °C, although to a very low extent. Both IgG and IgY were obtained by immunization with a total protein extract of (white) lupin seeds. Most probably, a proportion of the anti-bodies was raised against proteins with rather stable epitopes.

Reviewer 3 Report

Comments and Suggestions for Authors

This work compares different detection methods on lupin proteins after thermal processing. The workload is enough and the results are clearly presented. However, the design of the work lacks scientific soundness. The following are some comments:

1. It is true that lupin coffee involves thermal processing of lupin. However, design of this manuscript did not base on real industrial process. How did they prepare lupin seeds in practice? To produce the flavor compounds, MR is involved. MR for flavor and color depends not only on temperature but also on duration. For example, when the surface temperature reaches 242, the seeds became dark brown, but the same color and flavor may also be reached when holding at a lower temperature for longer duration.

2. For the extraction protocols, are they commonly used protocols in plant seed extraction? Where did you find the protocols, please give the citation or reason.

3. For differences of the two protocols, in the text it is explained that one had urea for denaturation but the other one didnt. Actually sonication for 15min also works on protein extraction. Please give corresponding explanation.

4. For line 386-394, mainly amino acids were used to explain why protein contents decreased. Actually during seed heating protein may also aggregate, would they reduce extraction rate?

5. In figure 1, add a tube for unheated lupin seed is better.

6. The conclusion part is too long, please pick out the important information.

7. Pay attention to the words in the text, for example, line 435, effects

Comments on the Quality of English Language

English proper,  minor change suggestions see comments and suggestions for authors

Author Response

We thank reviewer 3 for the helpful comments.

  1. It is true that “lupin coffee” involves thermal processing of lupin. However, design of this manuscript did not base on real industrial process. How did they prepare lupin seeds in practice? To produce the flavor compounds, MR is involved. MR for flavor and color depends not only on temperature but also on duration. For example, when the surface temperature reaches 242℃, the seeds became dark brown, but the same color and flavor may also be reached when holding at a lower temperature for longer duration.

We added information on heat treatment of lupin seeds designated for preparing lupin coffee.

In this study, we cooperated with a small-sized enterprise that is selling several tons of “lupin coffee” per year which is produced with the same roasting device that was used for this study. The roasting process was performed in a commercially available coffee roasting machine in a fluidized bed reactor with control of the temperature of the air stream as well as of the surface of the beans. For the commercial production of the lupin coffee, the roasting process is usually stopped at a surface temperature of the beans of 220°C; in this investigation, we prolonged the roasting process with the aim to follow the changes of the lupin seeds upon excess thermal load (please see also the changes in the manuscript). 

We agree with the reviewer that formation of MR products not only depends on temperature but also on duration. We have already given this information in the original version of the manuscript: “Melanoidin formation strongly depends on the type of food, the composition of amino acids and reducing sugars, heating temperature, and heating time [40].”  

  1. For the extraction protocols, are they commonly used protocols in plant seed extraction? Where did you find the protocols, please give the citation or reason.

As proposed by the reviewer, we added references.

Henrottin et al., 2023 presents recent outcomes of allergen detection by LC-MS/MS. The applied extraction protocol with urea and sonication has become the standard procedure for allergen detection and has frequently applied so far.

  1. For differences of the two protocols, in the text it is explained that one had urea for denaturation but the other one didn’t. Actually, sonication for 15min also works on protein extraction. Please give corresponding explanation.

We added the following information: “Additionally, the ultra-sonic treatment enhanced solubility due to disulfide bond breakage [44]…. Both extraction protocols reflect very well the individual circumstances/conditions of applied methodologies for protein quantification. On the one hand, mild conditions have to be applied for ELISA analyses to preserve protein structures; on the other side, for protein analysis by LC-MS/MS, extraction under reducing and denaturing conditions is used, commonly applied before digestion in order to improve breakdown of proteins to peptides as well.”

  1. For line 386-394, mainly amino acids were used to explain why protein contents decreased. Actually, during seed heating protein may also aggregate, would they reduce extraction rate?

We agree with the reviewer that aggregation also has an impact on protein extractability. We added this information as follows: “Heat treatment may also lead to protein aggregation, resulting in decreased protein solubility”.

  1. In figure 1, add a tube for unheated lupin seed is better.

As proposed by the reviewer, we included a Figure showing raw and heat treated lupin seeds.

  1. The conclusion part is too long, please pick out the important information.

As proposed by the reviewer, we shortened the conclusion; several parts of conclusion (in total four smaller paragraphs) have been removed.

  1. Pay attention to the words in the text, for example, line 435, “effects”

We carefully read the manuscript again and removed typing errors.

Reviewer 4 Report

Comments and Suggestions for Authors

1. Please explain the differences between lupin and lupin seed. (I want to know the species distribution of lupin allergen in lupin and lupin seed)

2. Whether the same calibration lupin allergen was used in ELISAs and LC-MS/MS test? If yes, please tell us its composition.

3. For LC-MS/MS test, please explain the principle of relative quantification method in detail (Not limited to data processing method).

4. Line 109, Why are these temperature gradients set?

5. Line 115, “ten” should be revised to “10”.

6. Line 150, Why did you choose non-linear regression?

Comments on the Quality of English Language

In this study, the author investigated the influence of roasting of seeds of the L. angustifolius cultivar “Boregine” on the detectability of lupin by (SDS-PAGE), ELISAs, LC-MS/MS, and real-time PCR. Seeds were roasted by fluidized bed roasting and samples were drawn at seed surface temperatures ranging from 98 °C to 242 °C. With increasing roasting temperature, the extractability of both proteins and DNA decreased. In addition, roasting resulted in lower detectability of lupin proteins by ELISAs and LC-MS/MS and lower detectability of DNA by real-time PCR. The whole story of this work is interesting, while there are some concerns which need to be revised in manuscript as I comments to the authors. 

Author Response

Reviewer 4

  1. Please explain the differences between lupin and lupin seed. (I want to know the species distribution of lupin allergen in lupin and lupin seed)

Since only seeds are relevant for food industry, all data on lupin allergens refer to lupin seeds and not to other parts of the lupin plant.

  1. Whether the same calibration lupin allergen was used in ELISAs and LC-MS/MS test? If yes, please tell us its composition.

In case of ELISAs, we used the protein extract obtained with Tris glycine, as outlined in the experimental selection. In case of LC-MS/MS, no calibration was applied and only relative abundance of proteins among modified (roasted) and control samples (raw) was compared.

  1. For LC-MS/MS test, please explain the principle of relative quantification method in detail (Not limited to data processing method).

    As proposed by the reviewer, we added further information on LC-MS/MS in the experimental section: “Separation and data acquisition was performed according to Geisslitz et al. [39]. SONAR mode with 35 Da window was applied for data independent acquisition (DIA). All samples were measured in triplicate. First, the raw data was processed in ProteinLynx Global server (PLGS) software (version 2.3) from WATERS via the Apex 3D algorithm for deconvolution and lock mass correction, followed by databank search queries with a fasta file containing 110 reviewed entries of lupine proteins downloaded on 9th December 2022. Label free quantification (LFQ) was performed based on exact mass retention time (EMRT) signatures generated for every eluting signal detected, followed by the expression analyses workflow of PLGS (raw lupin was assigned as “normal” and roasted samples as “modified”). The generated protein tables were manually checked for unique peptides based on the used fasta file for lupin. Peptides containing missed cleavage were removed and normalized intensities obtained from the expression workflow of PLGS were used to calculate relative intensities based on the untreated/raw lupin seeds, which was set as 1.0. The final calculations and heat maps were prepared in excel. Detailed setup of separation, data aquisition and processing are listed in the supplement (see Supplementary File).”

LFQ (label-free quantification) method is used for simultaneous identification and quantification of proteins. Compared with stable isotopic labeling approaches, LFQ is cost-effective and only minimal sample preparation is need. Furthermore, it also allows comparison across multiple experimental conditions such as in this study. In principle, LFQ methods are based on peptide intensities and their profiling. The relative abundance of proteins is calculated by the peak area of specific peptides, the share/relative abundance of unmodified and all modified forms. group/samples. To obtain reliable results three replicates should be measured. Additionally, normalization of PLGS software was applied to reduce variability of the method.

  1. Line 109, Why are these temperature gradients set?

    The lupin seeds were roasted in a professional pilot coffee roasting machine that is commercially used by a small-sized enterprise in the South of Austria. For these products, the seeds are usually roasted to a max. temperature of 220°. For this study, we decided to prolong the roasting process to follow the changes in the lupin seeds caused by excess thermal load. Sampling was started after drying of the seeds (surface temperature 98°C); further samples were drawn in temperature intervals of approx. 20°C to monitor the changes of the seeds in this temperature range.
  2. Line 115, “ten” should be revised to “10”.

We replaced ten by 10.0.

  1. Line 150, Why did you choose non-linear regression?

Calibration curves of ELISAs are not linear, but shows a sigmoidal shape (see Figure 4).  We therefore performed non-linear regression.

Reviewer 5 Report

Comments and Suggestions for Authors

Page 2, line 81: The rate of studies from recent years will increase by adding current ones to the references given for detecting Lupine by the ELISA method. Below are some of my suggestions:

Günay, N., Demirhan, B. E., Demirhan, B. (2022). Investigation of Lupine Allergen Presence in Some Food Products by Enzyme-Linked Immunosorbent Assay. Bulletin Of University Of Agricultural Sciences And Veterinary Medicine Cluj-Napoca-Food Science And Technology. 79(1): 36-40. https://doi.org/10.15835/buasvmcn-fst:2022.0006

Lima-Cabello, E., Alché, J. D., & Jimenez-Lopez, J. C. (2019). Narrow-leafed lupine main allergen β-conglutin (Lup an 1) detection and quantification assessment in natural and processed foods. Foods, 8(10), 513.

Page 3, line 131: Coomassie Brilliant Blue R-250 brand catalog number information should be added.

Page 4, line 163: Are Tris-HCl and urea molar solution or normal solution?

Page 5, line 210: “L. angustifolius L., cultivar “Boregine” The abbreviation L., used to indicate Linnaeus as the authority on the name of a species, should not be italicized.

Page 5, line 215: In the results, two methods for protein extraction were mentioned, but no information was given in the materials and methods section.

Page 3, line 118: (0.1 M Tris/0.5 M glycine, pH 8.7) and Page 5, line 215, 216: 0.5 M Tris, 0.1 M glycine, pH 8.7: Molarity of tris and glycine were different in the materials and methods and result.

In the protein extraction, figure 2b should have mentioned in the text.

Page 6, line 243: “Bands at ~85 kDa, ~68 kDa, and ~59 kDa” in the figure 3a I did not see these bands, either the data should be checked or these bands should be shown in the figure 3a.

Author Response

Reviewer 5

Page 2, line 81: The rate of studies from recent years will increase by adding current ones to the references given for detecting Lupine by the ELISA method. Below are some of my suggestions:

Günay, N., Demirhan, B. E., Demirhan, B. (2022). Investigation of Lupine Allergen Presence in Some Food Products by Enzyme-Linked Immunosorbent Assay. Bulletin Of University Of Agricultural Sciences And Veterinary Medicine Cluj-Napoca-Food Science And Technology. 79(1): 36-40. https://doi.org/10.15835/buasvmcn-fst:2022.0006

Lima-Cabello, E., Alché, J. D., & Jimenez-Lopez, J. C. (2019). Narrow-leafed lupine main allergen β-conglutin (Lup an 1) detection and quantification assessment in natural and processed foods. Foods, 8(10), 513.

We added the second references as proposed by the reviewer.

Page 3, line 131: Coomassie Brilliant Blue R-250 brand catalog number information should be added.

We do not agree with the reviewer that the brand catalogue number is necessary.

Page 4, line 163: Are Tris-HCl and urea molar solution or normal solution?

We checked the data, they are correct.

Page 5, line 210: “L. angustifolius L., cultivar “Boregine” The abbreviation L., used to indicate Linnaeus as the authority on the name of a species, should not be italicized.

Many thanks for the hint, we made the correction.

Page 5, line 215: In the results, two methods for protein extraction were mentioned, but no information was given in the materials and methods section.

The experimental details are given in sections 2.2.1 and 2.2.4.

Page 3, line 118: (0.1 M Tris/0.5 M glycine, pH 8.7) and Page 5, line 215, 216: 0.5 M Tris, 0.1 M glycine, pH 8.7: Molarity of tris and glycine were different in the materials and methods and result.

In the protein extraction, figure 2b should have mentioned in the text.

Many thanks for the hint, we corrected the mistake. 0.1 M Tris/0.5 M glycine, pH 8.7 is correct.

Page 6, line 243: “Bands at ~85 kDa, ~68 kDa, and ~59 kDa” in the figure 3a I did not see these bands, either the data should be checked or these bands should be shown in the figure 3a.

The molecular sizes are correct. They were calculated from the calibration equation obtained from the standard proteins.

Round 2

Reviewer 1 Report

Comments and Suggestions for Authors

In reviewing the revised manuscript, it is apparent that the authors have made substantial modifications and have diligently addressed the concerns raised by the reviewers, particularly in elucidating the effects of roasting on allergens. This revised version demonstrates a commendable level of thoroughness and is deemed suitable for publication in an academic context.

Author Response

We thank the reviewer for the helpful comments.

Reviewer 3 Report

Comments and Suggestions for Authors

Now the manuscript is qualified to publish

Author Response

(The authors gave the same response as above.)

Reviewer 5 Report

Comments and Suggestions for Authors

The article is suitable for publishing in its corrected form.

Author Response

(The authors gave the same response as above.)
